# Lifestyle risk behavior and atherosclerotic cardiovascular risk: An analysis using the Korea National Health and Nutrition Examination Survey

Minwoo Lee[1], Hyo-Jeong Ahn[2], Su Jung Lee[3], Pum-Jun Kim[4], Chulho Kim[5,6]*, Sang-Hwa Lee[5,6], Jong-Hee Sohn[5,6], Jae-Jun Lee[6,7]

1 Department of Neurology, Hallym University Sacred Heart Hospital, Anyang, Korea, 2 Health Insurance Review and Assessment Research Institute, Health Insurance Review & Assessment Service, Wonju, Korea, 3 Research Institute on Nursing Science, School of Nursing, Hallym University, Chuncheon, Korea, 4 Department of Artificial Intelligence, Ulsan National Institute of Science and Technology, Ulsan, Korea, 5 Department of Neurology, Chuncheon Sacred Heart Hospital, Chuncheon, Korea, 6 Institute of New Frontier Research Team, Hallym University College of Medicine, Chuncheon, Korea, 7 Department of Anaesthesiology and Pain Medicine, Chuncheon Sacred Heart Hospital, Chuncheon, Korea

* gumdol52@naver.com

**Data Availability Statement:** We used the Korean National Health and Nutrition Examination Survey (KNHANES) data, which could be publicly assessed from the website (https://knhanes.kdca.go.kr/

## Abstract

### Background

Clustering lifestyle risk behaviors is important for predicting cardiovascular disease risk. However, it is unclear which behavior mediates other ones to influence cardiovascular disease risk. We aimed to assess the causal inference of each lifestyle risk behavior for the atherosclerotic cardiovascular disease (ASCVD) risk of the general population.

### Methods

We performed a Bayesian network mediation analysis using data from the Korea National Health and Nutrition Examination Survey from 2014 to 2019. The main exposure was a combination of lifestyle risk behaviors including unhealthy weight, heavy alcohol consumption, inadequate sleep, physical inactivity, excessive sodium intake, and current smoking among subjects 40 to 79 years of age. The high risk of ASCVD ($\geq$7.5% for the 10-year risk) was assessed using logistic regression, Bayesian networks, and structural equational models to examine the causal relationships between these six lifestyle risk behaviors.

### Results

Among all participants, the most prevalent lifestyle risk behavior for those at high risk for ASCVD was excessive sodium intake (95.6%), followed by inadequate sleep (49.9%) and physical inactivity (43.8%). Older age (65–79 years) and male sex were directly associated with a high risk for ASCVD. Physical inactivity, current smoking, excessive sodium intake, and unhealthy weight indirectly mediated the effects of older age (8.2% of the older age) and male sex (39.9% of males) to high ASCVD risk. Physical inactivity, current smoking,

knhanes/sub03/sub03_02_05.do). Our data is available to our IRB with a reasonable request (chuncheonirb@hallym.or.kr).

**Funding:** This research was supported by the Ministry of Education / "Regional Innovation Strategy (RIS)" through the National Research Foundation of Korea (NRF, 2022RIS-005, awarded to CK), and by the Ministry of Health and Welfare / the Korea Health Technology R&D Project through the Korea Health Industry Development Institute (KHIDI, HR21C0198, awarded to JJL). The funders had no role in study design, data collection and analysis, decision to publish, or preparation of the manuscript.

**Competing interests:** The authors have declared that no competing interests exist.

excessive sodium intake, and unhealthy weight particularly mediated the high ASCVD risk sequentially. Heavy alcohol consumption and inadequate sleep were not directly associated with high ASCVD risk and did not indirectly mediate the effects of older age and males on the high ASCVD risk.

## Conclusion

Lifestyle risk behaviors mediated the atherosclerotic cardiovascular disease risk in a different manner. Especially, physical inactivity preceded current smoking, excessive sodium intake, and unhealthy weight in relation to high ASCVD risk, and this causal relationship was different according to age and sex. Therefore, tailored strategies according to specific target populations may be needed to effectively reduce the high ASCVD risk.

## Introduction

Cardiovascular disease (CVD) is one of the leading causes of death worldwide [1, 2]. In 2019, approximately 18.6 million people died from CVD [3], thus leading to a greater burden on global health than any other chronic disease [4]. Although the age-standardized mortality rates of CVD are consistently decreasing, the crude CVD mortality rates are increasing continuously because of global aging, advancements in the primary prevention of CVD, and changes in lifestyle and living environment [5].

To quantify and manage the CVD risk, atherosclerotic cardiovascular disease (ASCVD) risk estimation has been widely used during the past decades [6]. The primary method of minimizing the future risk of ASCVD is improving the lifestyle by modulating risk factors, especially smoking, unhealthy dietary patterns, and physical inactivity [7]. More than 50% of individuals at high risk for ASCVD have multiple lifestyle risk behaviors (LRB) [8, 9]. Recently, proper control of LRBs has been reported to effectively prevent recurrent cardiovascular events and reduce cardiovascular mortality [10]. Many prospective cohort studies have reported that the sum of LRBs affects CVD outcomes [7, 11]; however, knowledge of how each LRB affects the ASCVD risk is limited [12].

The evaluation of combinations of LRBs associated with CVD risk and preventive medicine is important because LRBs are not non-modifiable risk factors such as age and sex. Important modifiable risk factors can be sufficiently corrected to reduce CVD risk [13, 14]. Additionally, assessing associations between each LRB and risk is important for targeting which LRB should be corrected to effectively reduce the risk of specific diseases [15]. In other words, there may be a causal relationship between lifestyle factors and increased CVD risk. For example, deciding whether to correct smoking or control body weight is important for the intensive implementation of primary CVD prevention. Therefore, this study aimed to analyze the mediating effects of LRBs on the ASCVD risk and determine how LRBs influence each other in the general population.

## Methods

### Study population

During this study, we used data from the Korean National Health and Nutrition Examination Survey (KNHANES) conducted annually by the Division of Chronic Disease Surveillance under the guidance of the Korea Centers for Disease Control. The KNHANES is a nationwide

surveillance of data designed to evaluate and develop the health status and nutrition status of the representative Korean population through questionnaires including health interviews, nutrition surveys, and details of health examinations collected by professionally trained staff. Additionally, the survey applied a stratified, multistage, and probability sampling method to collect a representative sample of the study population.

This is the retrospective cross-sectional study to assess the relationship of each LRBs using network mediation analysis. We gathered and analyzed data from 2014 to 2019 in the KNHANES database. We included 25,639 participants 40 to 79 years of age so that the 10-year ASCVD risk score could be calculated. Among these data, 7309 observations, incomplete answers in the survey, and incomplete physical examination results were excluded (Fig 1).

## ASCVD risk estimation

The 10-year risk of ASCVD was estimated using clinical factors, demographic factors, and laboratory results, as defined by the American College of Cardiology/American Heart Association [6]. The estimate function of the ASCVD risk score for the white population was applied during this study because the estimation based on the white race had been widely used for other races in previous studies [16, 17]. The primary outcome was the estimated 10-year ASCVD risk score of more than 7.5% [18].

## Assessment of lifestyle risk behaviors

Because the methodologies of our analyses required different types of predictors, the categorical and continuous transformations of variables were conducted as appropriate. Body mass index (BMI) was calculated as weight in kilograms divided by the square of height in meters (kg/m$^2$) according to Asian Pacific World Health Organization criteria [19]. Unhealthy weight was defined as a BMI less than 18.5 (kg/m$^2$) or more than 25 (kg/m$^2$) [20]. Heavy alcohol consumption was defined as drinking 14 or more alcoholic beverages per week for men and 10 or more alcoholic beverages per week for women [21]. Current smoking was defined as smoking more than five packs of cigarettes during their lifetime and currently smoking at the time of the survey [21, 22]. Physical activity was calculated using the sum of the minutes of exercise per week and considering the strength of the exercise intensity (vigorous and moderate). The combination of vigorous-intensity and moderate-intensity exercise was considered by calculating the minutes exercised per week as follows: 2 × moderate activity (min/week) + vigorous activity (min/week). Physical inactivity was defined as performing less than 150 minutes of moderate-intensity physical activity per week [23]. We measured sleep duration by checking the number of hours of sleep per week; we defined inadequate sleep as less than 7 hours or 9 hours or more [24]. Sodium intake was estimated using Tanaka's equation using the measured amount of spot urine sodium (mmol/L) and creatine (mg/dL) per 24 hours. Excessive sodium intake was defined as more than 87 mmol of urine sodium excretion over the course of 24 hours [25]. Age was dichotomized into ages 40 to 64 years and 65 to 79 years. Additional definitions of conventional cardiovascular risk factors and demographic variables are presented in the Supplementary material online (S1 Text).

## Statistical analysis

We assessed the baseline differences of independent variables of dichotomized ASCVD risk score groups (≥7.5% vs. <7.5%) using the $\chi^2$-test and Student's t-test. We used the following two-step statistical approach to assess how each LRB or any correlation of LRBs affects the ASCVD risk. First, a binary logistic regression analysis was performed to classify the subjects with high ASCVD risk scores (≥7.5%) and to check whether a significant covariance shift

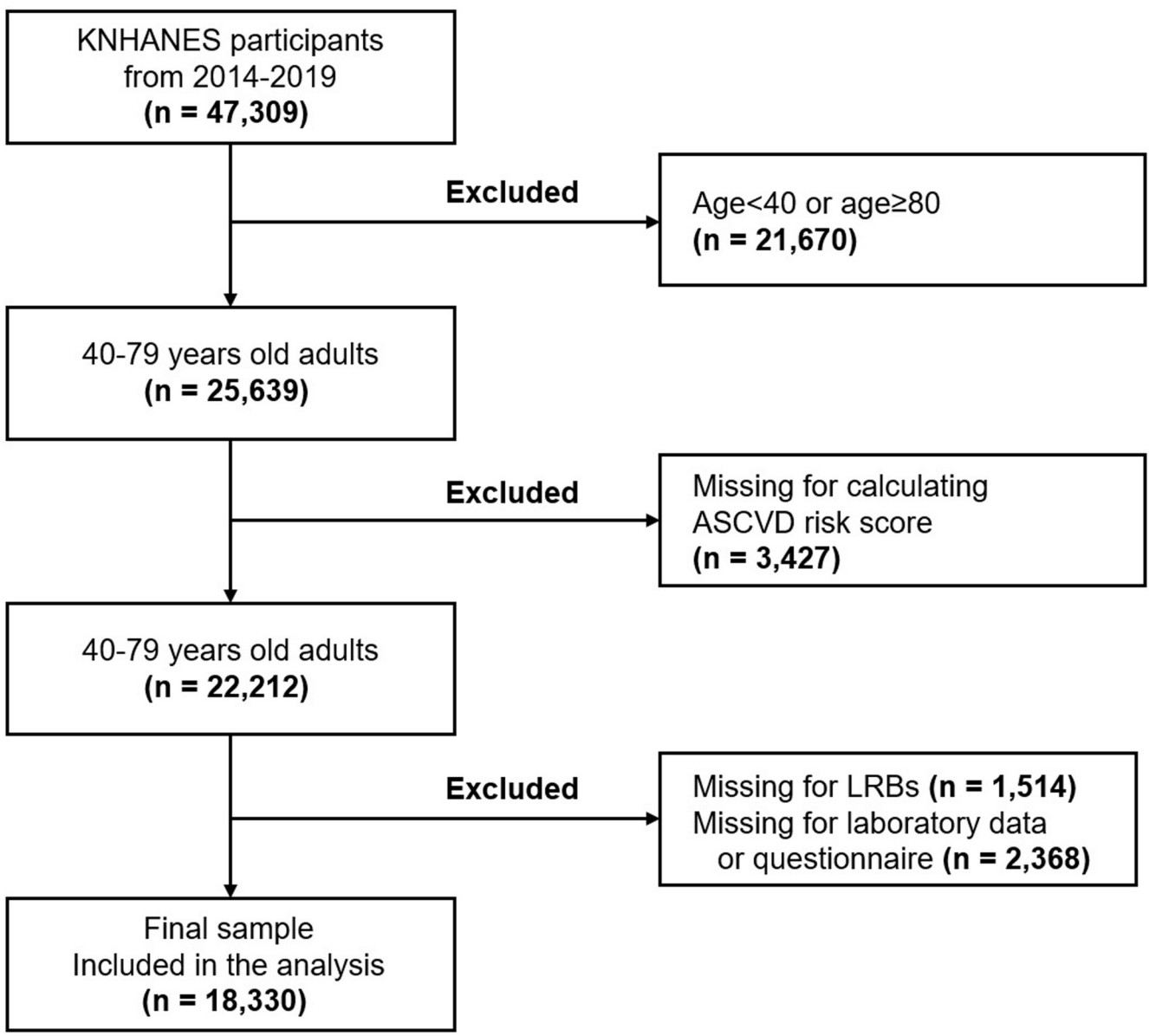

**Fig 1. Flowchart of the subject selections from the KNHANES database.**

existed when using categorized predictors in the model. As stated in the introduction, we set the outcome to be the ASCVD risk, not the development of the ASCVD itself. Therefore, we did not include vascular risk factors in logistic regression analysis because if vascular risk factors were entered into the model to determine the association between the ASCVD risk and lifestyle risk behaviors, it would only reflect the importance of variables directly reflected in ASCVD risks, such as age, gender, hypertension, diabetes, and dyslipidemia.

Second, we investigated the causal effects of LRBs on the ASCVD risk using the Bayesian network mediation analysis and structural equation model (SEM). Bayesian network mediation is a statistical method for investigating causal relationships between variables in a dataset. Specifically, it allows researchers to determine whether the relationship between two variables is mediated by one or more additional variables. A Bayesian network is a probabilistic

graphical model that represents the relationships between variables in a dataset. In a Bayesian network, nodes represent variables and edges represent conditional dependencies between variables by a directed acyclic graph (DAG). A Markov Blanket is a set of variables that contains all of the variables that are direct causes, direct effects, or direct confounders of a given variable [26]. We used a constraint-based algorithm called the grow-shrink Markov blanket to learn the Bayesian network model structure [27]. The bootstrap approach was repeated 200 times, and an average DAG representing the final Bayesian network was drawn. We used sex and age as prior nodes (non-modifiable variables) of conditional dependency in the Bayesian network model and combinations of LRBs (modifiable variables) as mediators of probabilistic inference of the ASCVD risk. The detailed procedure for the Bayesian network mediation analysis was summarized in the Supplementary material online (S2 Text). Additionally, we used the SEM model to quantify the direct effects of age and sex on the ASCVD risk and their indirect effects through LRBs. We used the $R^2$ value for the explanatory power of the SEM model. All analyses were performed using the Bnlearn (version 4.7) and Lavaan (version 0.6–9) packages for R software (R Foundation for Statistical Computing version 4.0.3).

### Ethics statement

This study was approved by the Institutional Review Board of Chuncheon Sacred Heart Hospital (IRB no. 2021-12-006), and the need for informed consent was waived because we used the fully deidentified public database.

### Results

This study included 18,330 participants (mean age, 58.3±10.8 years; 41.8% males). The clinical and demographic characteristics of the predefined ASCVD risk score groups are demonstrated in Table 1. The proportion of participants with high ASCVD scores was 42.7% (7832/18,330). Age and male sex were positively associated with a high ASCVD risk. However, the academic year, household income level, white race, occupation, and being married were negatively associated with the ASCVD risk. The most prevalent LRBs in the high ASCVD risk group were excessive sodium intake (95.6%), followed by inadequate sleep (49.9%), physical inactivity (43.8%), and unhealthy weight (43.3%).

The association between each LRB and the risk of having a high ASCVD score (>7.5%) using binary logistic regression analyses is demonstrated in Table 2. All six LRBs were statistically significant factors in the univariate analyses. Among them, current smoking (odds ratio [OR], 3.73; 95% confidence interval [CI], 3.31–4.22), unhealthy weight (OR, 1.51; 95% CI; 1.37–1.66), and inadequate sleep (OR, 1.11; 95% CI, 1.01–1.22) remained significant in the multivariate analysis after adjusting for age, sex, educational status, occupation, household income, and marital status.

Fig 2 shows the results of the Bayesian network model of the association of LRBs and high ASCVD risk, which provide probabilistic inference pathways that could be mediated by age, sex, and several combinations of LRBs. In this network model, older age (65–79 years) and male sex were directly associated with a high ASCVD risk, and physical inactivity, current smoking, excessive sodium intake, and unhealthy weight indirectly mediated the effects of older age and male sex to a high ASCVD risk. The older age group was positively associated with unhealthy weight and negatively associated with current smoking. Male sex was positively associated with current smoking, excessive sodium intake, unhealthy weight, and heavy alcohol consumption, and it was negatively associated with physical inactivity. Excessive sodium intake was not directly associated with a high ASCVD risk; however, it was indirectly associated with ASCVD only through the mediation of unhealthy weight. Heavy alcohol

**Table 1. Baseline characteristics of lifestyle behavioral risk factors associated with ASCVD risk scores.**

|  | ASCVD Score <7.5% (N = 10498) | ASCVD Score ≥7.5% (N = 7832) | p |
|---|---|---|---|
| Male | 2879 (27.4%) | 4806 (61.4%) | <0.001 |
| Age, years |  |  | <0.001 |
| 40–64 | 9946 (94.7%) | 2583 (33.0%) |  |
| 65–79 | 552 (5.3%) | 5249 (67.0%) |  |
| Educational year |  |  | <0.001 |
| <6 | 1301 (12.4%) | 3452 (44.1%) |  |
| 6–9 | 1251 (11.9%) | 1273 (16.2%) |  |
| 9–12 | 3954 (37.7%) | 1837 (23.5%) |  |
| ≥12 | 3992 (38.0%) | 1270 (16.2%) |  |
| Level of income |  |  | 0.009 |
| Low | 2384 (22.7%) | 1911 (24.4%) |  |
| Mid-low | 2629 (25.1%) | 1959 (25.0%) |  |
| Mid-high | 2680 (25.5%) | 2015 (25.7%) |  |
| High | 2805 (26.7%) | 1947 (24.9%) |  |
| Occupation |  |  | <0.001 |
| White collar | 4693 (44.7%) | 1255 (16.0%) |  |
| Blue collar | 2507 (23.9%) | 2652 (33.9%) |  |
| Unemployed | 3298 (31.4%) | 3925 (50.1%) |  |
| Married | 5419 (84.9%) | 4217 (75.4%) | <0.001 |
| Unhealthy weight | 3654 (34.8%) | 3394 (43.3%) | <0.001 |
| Heavy alcohol consumption | 936 (8.9%) | 881 (11.2%) | <0.001 |
| Inadequate sleep | 4525 (43.1%) | 3910 (49.9%) | <0.001 |
| Physical inactivity | 3927 (37.4%) | 3430 (43.8%) | <0.001 |
| Excessive sodium intake | 9963 (94.9%) | 7486 (95.6%) | 0.037 |
| Current smoking | 967 (9.2%) | 1798 (23.0%) | <0.001 |

The p-values of the $\chi^2$ test represent the statistical significance of categorical variables. ASCVD, atherosclerotic cardiovascular disease; BMI, body mass index.

consumption and inadequate sleep were not directly associated with high ASCVD risk and did not indirectly mediate the effects of older age and male sex to the high ASCVD risk. Additionally, physical inactivity, current smoking, excessive sodium intake, and unhealthy weight were

**Table 2. Univariate and multivariate logistic regression analyses results of lifestyle risk behaviors and high ASCVD risk scores.**

|  | OR (95% CI) | | | |
|---|---|---|---|---|
|  | Crude Model | Model 1 | Model 2 | Model 3 |
| Unhealthy weight | 1.43 (1.35–1.52) | 1.40 (1.32–1.49) | 1.50 (1.37–1.64) | 1.51 (1.37–1.66) |
| Heavy alcohol consumption | 1.29 (1.17–1.43) | 0.91 (0.82–1.01) | 1.02 (0.90–1.17) | 1.01 (0.88–1.16) |
| Inadequate sleep | 1.32 (1.24–1.40) | 1.31 (1.23–1.39) | 1.19 (1.09–1.31) | 1.11 (1.01–1.22) |
| Physical inactivity | 1.30 (1.23–1.38) | 1.25 (1.17–1.32) | 1.26 (1.15–1.38) | 1.09 (0.99–1.21) |
| Excessive sodium intake | 1.16 (1.01–1.34) | 1.22 (1.06–1.41) | 1.01 (0.81–1.25) | 1.09 (0.87–1.37) |
| Current smoking | 2.94 (2.70–3.20) | 2.95 (2.71–3.23) | 3.58 (3.20–4.02) | 3.73 (3.31–4.22) |

Crude model: lifestyle risk behaviors 1–6 are considered univariate predictors. Model 1: lifestyle risk behaviors 1–6 are considered multivariate predictors. Model 2: lifestyle risk behaviors 1–6 are considered multivariate predictors after adjusting for age and sex. Model 3: lifestyle risk behaviors 1–6 are considered multivariate predictors after adjusting for age, sex, occupation, household income, and marital status.

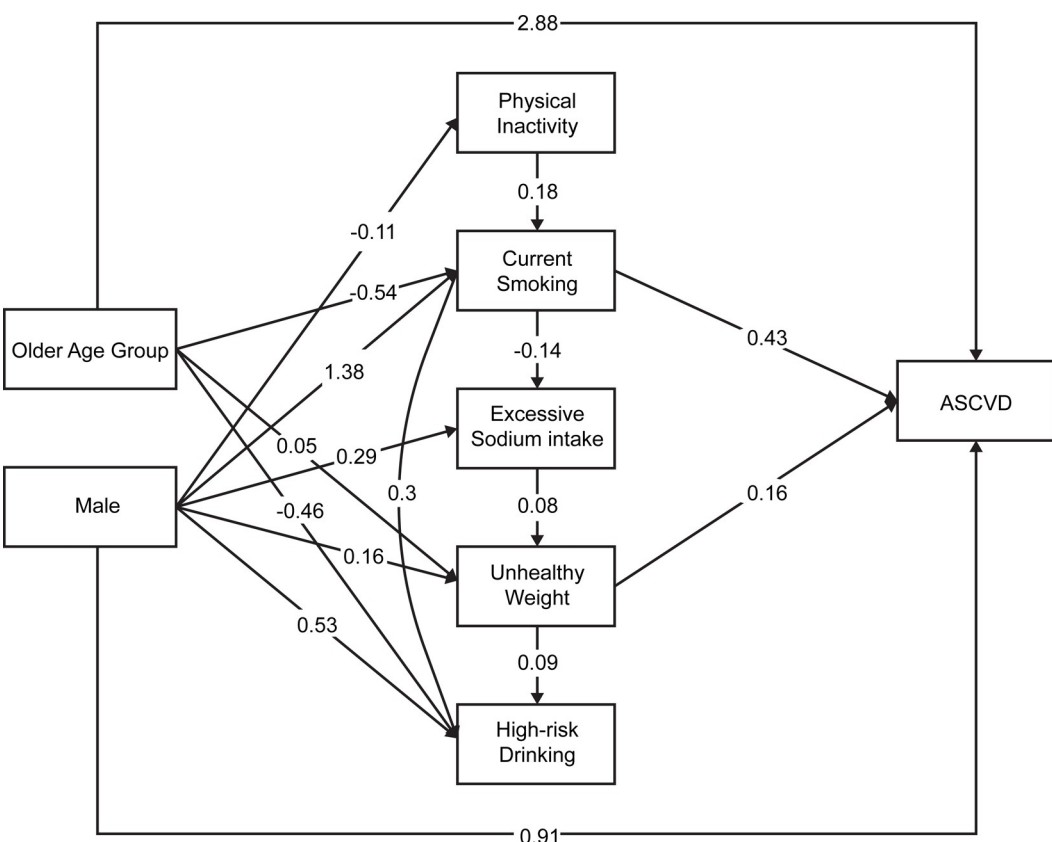

**Fig 2. Path diagram of the relationship between lifestyle risk behaviors (mediators), and high ASCVD risk (outcome) in the presence of measured and unmeasured confounding factors.** ASCVD, atherosclerotic cardiovascular disease. In the model, age and older age are placed in antecedent because they are non-modifiable risk factors. Only the variables that enter the logistic regression analysis will enter into the grow-shrink Markov blanket to determine the predictor (antecedent, start point of the arrow) and the outcome (outcome, end point of the arrow).

shown to mediate the high ASCVD risk sequentially. Additionally, excessive sodium intake negatively mediated the effect of current smoking on the high ASCVD risk.

Table 3 demonstrates the indirect effects of LRBs on developing high risk for ASCVD in the SEM model. The overall paths indicated the significant indirect effects of physical inactivity,

**Table 3. Paths of mediation analyses examining indirect effects of lifestyle risk behaviors on high ASCVD scores.**

| Independent Variable | Indirect Effect | | | | |
|---|---|---|---|---|---|
| | | Path | β | SE | p |
| Older age group (65–79 years) | → | CS → ASCVD | -0.225 | 0.015 | <0.001 |
| | → | UW → ASCVD | 0.008 | 0.003 | 0.010 |
| Male | → | CS → ASCVD | 0.598 | 0.026 | <0.001 |
| | → | PI → CS → ASCVD | -0.008 | 0.002 | <0.001 |
| | → | UW → ASCVD | 0.023 | 0.004 | <0.001 |
| | → | ES → UW → ASCVD | 0.004 | 0.001 | 0.001 |
| | → | CS → ES → UW → ASCVD | -0.002 | 0.001 | 0.001 |

CS = current smoking; IS = inadequate sleep; HAC = heavy alcohol consumption; UW = unhealthy weight; PI = physical inactivity; ES = excessive sodium intake; ASCVD, atherosclerotic cardiovascular disease; β, standardized regression coefficient; SE, standard error.

excessive sodium intake, current smoking, and unhealthy weight on the high ASCVD risk. The direct effects of older age and the male sex mainly accounted for the development of the high ASCVD risk. However, indirect mediation by LRBs explained 8.2% and 39.9% of the direct effects of older age and male sex, respectively (Table 4). The $R^2$ value for the SEM model was 0.75, which indicated a high percentage of the variance of the ASCVD risk explained by older age, male sex, and LRBs.

## Discussion

During this study, we examined the causal inference of each LRB with the ASCVD risk in the general population and demonstrated that nearly half of the participants had a 10-year ASCVD risk of 7.5% or more. Moreover, high-risk LRBs were highly prevalent in this general population. Excessive sodium intake was the most prevalent LRB, followed by inadequate sleep, physical inactivity, and unhealthy weight. In our Bayesian network model and SEM used to assess the mediation effects of LRBs, old age, and male sex accounted for most of the ASCVD risk. However, physical inactivity increased the ASCVD risk by mediating current smoking, excessive sodium intake, and unhealthy weight. The direct effects of old age and male sex on the ASCVD risk represented most of the risk, but the indirect effects of LRBs were found to be statistically significant mediators of increasing the ASCVD risk. To reduce the additional risk of ASCVD mediated by LRB, focusing on physical inactivity and current smoking, rather than reducing excessive sodium intake and unhealthy weight, as targets is an effective prevention method.

Lifestyle modifications, including smoking cessation and maintaining a high level of physical activity, are effective preventive measures that can reduce the CVD risk and maintain overall health. As described, many studies have reported that the CVD risk can be reduced by correcting multiple LRBs. However, it is clinically relevant to evaluate which of the LRBs precedes the effect modifier to reduce the CVD risk using limited preventive resources. To our best knowledge, only a few studies have reported the causal relationship between several LRBs and CVD risk in a prospective cohort. Although this study used cross-sectional and retrospective data, the Bayesian network model was used to present which LRB component can probabilistically precede the other components.

According to our results, physical inactivity preceded current smoking and was shown to affect the high ASCVD risk of the male subjects. Results of a systematic review of physical activity for smoking cessation showed that exercise intervention did not have sufficient preventive power to stop smoking in 20 clinical trial populations [28]. However, the small sample size of these trials raised the possibility that intense physical exercise intervention may be sufficiently effective for smoking cessation [28]. According to a previous study, a low level of physical activity was associated with depression and physical dependency on nicotine [29]. Another study also supported the evidence that physical activity reduces nicotine cravings and withdrawal symptoms, which in turn, cause successful smoking cessation [30]. A negative impact of cigarette smoking on appetite (sodium intake) has been reported by many experimental and

**Table 4. Results of mediation analyses with the mediated proportions of lifestyle risk behaviors.**

| Independent Variable | Direct Effect | | | Total Indirect Effect | | | Total Effect | | | Mediated Proportion |
|---|---|---|---|---|---|---|---|---|---|---|
| | β | SE | p | β | SE | p | β | SE | p | % |
| Older age group (65–79 years) | 2.875 | 0.037 | <0.001 | -0.217 | 0.016 | <0.001 | 2.658 | 0.035 | <0.001 | 8.2% |
| Male | 0.922 | 0.038 | <0.001 | 0.229 | 0.024 | <0.001 | 1.150 | 0.043 | <0.001 | 39.9% |

The mediated proportion is presented as the total indirect effect/total effect. β, standardized regression coefficient; SE, standard error.

clinical studies [31, 32]. Besides, heavy alcohol consumption was not a modulator of the high ASCVD according to our results. However, a prospective Russian cohort study showed that frequent heavy alcohol consumption (alcohol intake ≥80 g/day and ≥3 times/week) was associated with a two-fold increase in CVD mortality [33]. An insufficient adjustment of confounders, including LRBs and frequent heavy alcohol consumption episodes during this study, might partially explain the weak or absence of associations between heavy alcohol consumption and high ASCVD risk shown by our data. However, a dose-response relationship between alcohol consumption and CVD risk is an undeniable truth in the usual setting.

In our Bayesian network model, antecedent current smoking was associated with consequent excessive sodium intake in a negative manner (Fig 2). There are several studies that smoking has an appetite-lowering effect. Increasing oral nicotine intake gradually decreased food intake in a clinical trial [34]. Moreover, adult smokers tend to have a lower BMI and unhealthy eating habits compared to non-smokers, and smoking cessation was associated with weight gain [35]. The results of our Bayesian network graph suggesting that smoking had a negative effect on excessive sodium intake are consistent with these reports. Bayesian network mediation analysis using the grow-shrink Markov Blanket is a powerful method that can help researchers identify causal pathways between variables in a dataset.

According to our data, the male sex was directly and indirectly associated with a high ASCVD risk. Vyssoulis et al. analyzed the CVD risk factor profiles of 21,280 Greek patients with hypertension [36]. Of those subjects, smoking, diabetes, and high triglyceride concentrations were more prevalent in men, and these differences were more evident in young subjects. According to the European Action on Secondary and Primary Prevention through Intervention to Reduce Events survey that included 7998 patients with established coronary artery disease from 24 European Union countries, clustering of cardiovascular risk factors such as smoking, obesity, high blood pressure, high low-density lipoprotein, and diabetes were more prevalent in women than in men [37]. The high prevalence of the CVD risk factor profile in this survey could be explained by the slightly higher proportion of older women than older men (39.1% vs. 27.3%). In addition to this age gap between men and women, they suggested that sex differences in CVD risk factor profiles could be explained by residual confounders such as educational status, target achievement for risk factor control, and LRBs [37]. According to our data, male subjects were more likely to be associated with unhealthy LRBs than women. Therefore, our data support the hypothesis that LRBs could be confounders for this sex difference in ASCVD risk estimation.

We used the Bayesian network and the SEM models to assess the causal relationship of each LRB with high ASCVD risk using cross-sectional data. The Bayesian network can simultaneously evaluate the relationships between one or more variables, including dependency and independency. With multilevel models, the covariance estimation of random probability effects involves two mediation regression equations with different dependent variables. Therefore, we obtained estimates of the coefficients in regression models, and the resulting DAG was interpreted as causal nodes of relationships. Therefore, the Bayesian network can measure causal relationships of multiple factors and effectively identify the effect modifiers and confounders with complex pathways. Although there is no universally accepted method of constructing the Bayesian network from data, an additional SEM could effectively assess the interactions of the mediator variables, as in our study.

Our study had several limitations. First, the LRB information was not prospectively collected; therefore, there might have been significant sectional bias because of the unmeasured confounders. However, we sufficiently adjusted for demographic and socioeconomic factors. Second, we only assessed the ASCVD risk and not actual ASCVD events. While ASCVD risk is an important predictor of future cardiovascular events, it is important to acknowledge that

actual events may differ from predicted risk. Therefore, the effects of LRBs on the ASCVD risk might be different compared to other clinical studies using real CVD outcomes.

Despite these limitations, our study had several strengths. First, we used nationally representative data, and complex multistage sampling was performed to reflect the fundamental demographic characteristics in Korea. Second, the advantage of using KNHANES data is that all participants were of a single ethnic group; therefore, we did not have to consider the effects of multi-ethnicity on LRBs when conducting this study. Finally, the dietary intake level (e.g., excessive sodium intake) was quantified by a laboratory test instead of a dietary questionnaire.

In conclusion, this nationally representative population-based study revealed that highly prevalent LRBs were directly and indirectly associated with high ASCVD risk. Using Bayesian network mediation and the SEM models, we identified that physical inactivity might precede current smoking, excessive sodium intake, and unhealthy weight. Although most ASCVD risks were directly caused by old age and male sex, important mediators, including physical inactivity, current smoking, excessive sodium intake, and unhealthy weight, were also identified as high ASCVD risk factors. Our results provide evidence that LRBs mediate ASCVD risks in the Bayesian network model.

## Supporting information

**S1 Text. Definitions for additional covariates.**
(DOCX)

**S2 Text. A detailed explanation of the Bayesian network mediation analysis.**
(DOCX)

## Author Contributions

**Conceptualization:** Minwoo Lee, Su Jung Lee, Chulho Kim, Jong-Hee Sohn, Jae-Jun Lee.

**Data curation:** Hyo-Jeong Ahn, Pum-Jun Kim.

**Formal analysis:** Hyo-Jeong Ahn, Pum-Jun Kim.

**Investigation:** Minwoo Lee.

**Methodology:** Su Jung Lee, Jae-Jun Lee.

**Resources:** Jae-Jun Lee.

**Software:** Sang-Hwa Lee.

**Supervision:** Chulho Kim, Jong-Hee Sohn, Jae-Jun Lee.

**Validation:** Chulho Kim, Jong-Hee Sohn.

**Writing – original draft:** Minwoo Lee.

**Writing – review & editing:** Chulho Kim, Sang-Hwa Lee.

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
