## [Decision Letter · Decision Letter 0]

25 Jun 2024

PONE-D-24-16325Lifestyle Risk Behavior and Atherosclerotic Cardiovascular Risk: An Analysis Using the Korea National Health and Nutrition Examination SurveyPLOS ONE

Dear Dr. Kim,

Thank you for submitting your manuscript to PLOS ONE. After careful consideration, we feel that it has merit but does not fully meet PLOS ONE’s publication criteria as it currently stands. Therefore, we invite you to submit a revised version of the manuscript that addresses the points raised during the review process.

We look forward to receiving your revised manuscript.

Kind regards,

Hidetaka Hamasaki

Academic Editor

PLOS ONE

Journal Requirements:

Reviewers' comments:

Reviewer's Responses to Questions

**Comments to the Author**

1. Is the manuscript technically sound, and do the data support the conclusions?

Reviewer #1: Yes

Reviewer #2: Yes

2. Has the statistical analysis been performed appropriately and rigorously? 

Reviewer #1: I Don't Know

Reviewer #2: Yes

3. Have the authors made all data underlying the findings in their manuscript fully available?

Reviewer #1: Yes

Reviewer #2: Yes

4. Is the manuscript presented in an intelligible fashion and written in standard English?

Reviewer #1: Yes

Reviewer #2: Yes

5. Review Comments to the Author

Reviewer #1: Dear editor,

Thank you for allowing me to review this manuscript.

The authors present a manuscript with valuable insights into the relationship between lifestyle risk behaviors and ASCVD risk in the Korean population, employing robust statistical methods and utilizing nationally representative data from KNHANES.

Here are my minor comments:

- In the discussion section; the second paragraph lacks references. “Lifestyle modifications, including smoking cessation and maintaining a high level of physical activity, are effective preventive measures that can reduce the CVD risk and maintain overall health. As described, many studies have reported that the CVD risk can be reduced by correcting multiple LRBs. However, it is clinically relevant to evaluate which of the LRBs precedes the effect modifier to reduce the CVD risk using limited preventive resources. To our best knowledge, only a few studies have 9 reported the causal relationship between several LRBs and CVD risk in a prospective cohort. Although this study used cross-sectional and retrospective data, the Bayesian network model was used to present which LRB component can probabilistically precede the other components.”

- I encourage the authors to provide a paragraph on the discussion section addressing how can healthcare professionals utilize this information to improve CVD risk assessment and prevention strategies.

Reviewer #2: 1. The manuscript is technically sound, and the data supported the conclusions.

2. The statistical analysis has been performed appropriately and rigorously.

3. The authors have made all data underlying the findings in their manuscript fully available.

4. The manuscript is presented in an intelligible fashion and written in standard English.

6. PLOS authors have the option to publish the peer review history of their article (what does this mean?). If published, this will include your full peer review and any attached files.

Reviewer #1: No

Reviewer #2: **Yes: **Kassa Demissie Abdi (PhD)

---

## [Author Response · Author response to Decision Letter 0]

2 Jul 2024

MS ID#: PONE-D-24-16325

MS TITLE: Lifestyle Risk Behavior and Atherosclerotic Cardiovascular Risk: An Analysis Using the Korea National Health and Nutrition Examination Survey

Dear Dr. Hidetaka Hamasaki

Thank you for providing us with the opportunity to revise and resubmit our manuscript entitled ‘Lifestyle Risk Behavior and Atherosclerotic Cardiovascular Risk: An Analysis Using the Korea National Health and Nutrition Examination Survey’ (MS ID: PONE-D-24-16325) for possible publication in PLOS ONE. We are sincerely grateful to the reviewers for their insightful comments, which have significantly contributed to the enhancement of our manuscript.

Enclosed, please find our revised manuscript. We have provided point-by-point responses to the reviewers’ comments below. The reviewers’ feedback is highlighted in blue text and set in an 11-point font, while our response can be found directly beneath each reviewer’s comment. For your convenience, all changes made to the manuscript are highlighted with a yellow background.

It is our hope that, with these revisions, our manuscript will be deemed suitable for publication in PLOS ONE. Should you require any further information or clarification, please do not hesitate to contact me. 

Sincerely yours,

Chulho Kim, MD, PhD

Department of Neurology, Chuncheon Sacred Heart Hospital, Hallym University College of Medicine, 77 Sakju-ro, 24253 Chuncheon, Korea

Telephone: +82-33-240-5255

Fax: +82-33-255-1338

E-mail: gumdol52@naver.com

Reviewer #1

The authors present a manuscript with valuable insights into the relationship between lifestyle risk behaviors and ASCVD risk in the Korean population, employing robust statistical methods and utilizing nationally representative data from KNHANES. 

Here are my minor comments: 

Comment #1

In the discussion section; the second paragraph lacks references. “Lifestyle modifications, including smoking cessation and maintaining a high level of physical activity, are effective preventive measures that can reduce the CVD risk and maintain overall health. As described, many studies have reported that the CVD risk can be reduced by correcting multiple LRBs. However, it is clinically relevant to evaluate which of the LRBs precedes the effect modifier to reduce the CVD risk using limited preventive resources. To our best knowledge, only a few studies have 9 reported the causal relationship between several LRBs and CVD risk in a prospective cohort. Although this study used cross-sectional and retrospective data, the Bayesian network model was used to present which LRB component can probabilistically precede the other components.”

Response #1

We thank the reviewer for this insightful comment. To address this, we have cited appropriate articles we have reviewed while writing the manuscript in the paragraph.

Lifestyle modifications, including smoking cessation and maintaining a high level of physical activity, are effective preventive measures that can reduce the CVD risk and maintain overall health7,10,11. As described, many studies have reported that the CVD risk can be reduced by correcting multiple LRBs10,11. However, it is clinically relevant to evaluate which of the LRBs precedes the effect modifier to reduce the CVD risk using limited preventive resources. To the best of our knowledge, only a few studies have reported the causal relationship between several LRBs and CVD risk in a prospective cohort28.29. Although this study used cross-sectional and retrospective data, the Bayesian network model was used to present which LRB component can probabilistically precede the other components.

References

28.Liu G, Li Y, Hu Y, Zong G, Li S, Rimm EB, et al. Influence of lifestyle on incident cardiovascular disease and mortality in patients with diabetes mellitus. Journal of the American College of Cardiology. 2018;71:2867-2876

29.Nambo R, Karashima S, Mizoguchi R, Konishi S, Hashimoto A, Aono D, et al. Prediction and causal inference of cardiovascular and cerebrovascular diseases based on lifestyle questionnaires. Sci Rep. 2024;14:10492

Comment #2

I encourage the authors to provide a paragraph on the discussion section addressing how can healthcare professionals utilize this information to improve CVD risk assessment and prevention strategies. 

Response #2

We thank the reviewer for this valuable suggestion. In response, we have added a paragraph in the discussion section that addresses how healthcare professionals can utilize the findings of our study to improve CVD risk assessment and prevention strategies. The new paragraph has been included below:

[Discussion, Last paragraph]

Our study results can be utilized by healthcare professionals to promote healthy lifestyles for patients, with a particular focus on encouraging regular physical activity to mitigate sequential LRBs, especially among older males, given the significant role of LRBs in mediating ASCVD risk. Additionally, understanding the interplay between different LRBs allows for more personalized and effective preventive measures, ultimately optimizing resource allocation and enhancing patient outcomes.

Reviewer #2

Comment #1

1. The manuscript is technically sound, and the data supported the conclusions.

2. The statistical analysis has been performed appropriately and rigorously.

3. The authors have made all data underlying the findings in their manuscript fully available.

4. The manuscript is presented in an intelligible fashion and written in standard English.

Response #1 

We thank the reviewer for their positive comments and appreciation of our work.

---

## [Decision Letter · Decision Letter 1]

10 Jul 2024

Lifestyle Risk Behavior and Atherosclerotic Cardiovascular Risk: An Analysis Using the Korea National Health and Nutrition Examination Survey

PONE-D-24-16325R1

Dear Dr. Kim,

We’re pleased to inform you that your manuscript has been judged scientifically suitable for publication and will be formally accepted for publication once it meets all outstanding technical requirements.

Kind regards,

Hidetaka Hamasaki

Academic Editor

PLOS ONE

Additional Editor Comments (optional):

Reviewers' comments:

Reviewer's Responses to Questions

**Comments to the Author**

1. If the authors have adequately addressed your comments raised in a previous round of review and you feel that this manuscript is now acceptable for publication, you may indicate that here to bypass the “Comments to the Author” section, enter your conflict of interest statement in the “Confidential to Editor” section, and submit your "Accept" recommendation.

Reviewer #2: All comments have been addressed

2. Is the manuscript technically sound, and do the data support the conclusions?

Reviewer #2: Yes

3. Has the statistical analysis been performed appropriately and rigorously? 

Reviewer #2: Yes

4. Have the authors made all data underlying the findings in their manuscript fully available?

Reviewer #2: Yes

5. Is the manuscript presented in an intelligible fashion and written in standard English?

Reviewer #2: Yes

6. Review Comments to the Author

Reviewer #2: 1. The authors have adequately addressed my comments raised in a previous round of review and I feel that this manuscript is now acceptable for publication.

2. The manuscript is technically sound and the data support the conclusions.

3. The statistical analysis has been performed appropriately and rigorously.

4. The authors have made all data underlying the findings in their manuscript fully available.

5. The manuscript is presented in an intelligible fashion and written in standard English.

7. PLOS authors have the option to publish the peer review history of their article (what does this mean?). If published, this will include your full peer review and any attached files.

Reviewer #2: **Yes: **Kassa Demissie Abdi (PhD)

---

## [Editor Report · Acceptance letter]

18 Jul 2024

PONE-D-24-16325R1 

PLOS ONE

Dear Dr. Kim, 

I'm pleased to inform you that your manuscript has been deemed suitable for publication in PLOS ONE. Congratulations! Your manuscript is now being handed over to our production team.

Kind regards, 

on behalf of

Dr. Hidetaka Hamasaki 

Academic Editor

PLOS ONE